# Stereotactic Radiofrequency Ablation for Treatment-Refractory Depression: A Systematic Review and Meta-Analysis

**DOI:** 10.3390/brainsci12101379

**Published:** 2022-10-12

**Authors:** Pauline Sarah Münchenberg, Eileen M. Joyce, Keith Matthews, David Christmas, Ludvic Zrinzo

**Affiliations:** 1Department of Clinical & Motor Neurosciences, UCL Institute of Neurology, Queen Square, London WC1N 3BG, UK; 2Advanced Interventions Service, Ninewells Hospital and Medical School, Dundee DD1 9SY, UK

**Keywords:** major depressive disorder, stereotactic radiofrequency ablation, treatment-refractory depression, anterior cingulotomy, anterior capsulotomy

## Abstract

(1) Background: Major depressive disorder (MDD) generates a large proportion of global disease burden. Stereotactic radiofrequency ablation (SRA) may be beneficial for selected patients with its most debilitating and refractory forms, but effect size is uncertain. (2) Methods: A systematic literature review and meta-analysis on SRA for MDD was carried out. Patient-level data were extracted from articles reporting validated depression measures (Beck Depression Inventory (BDI), Montgomery–Åsberg Depression Rating Scale (MADRS)), pre- and at least six months post surgery. To accommodate different outcome measures, the standardised mean difference (SMD) between both scores was used as the principal effect size. Data were synthesised using a random-effects model. (3) Results: Five distinct studies were identified, comprising 116 patients (64 included in meta-analysis). Effect size comparing post- vs. pre-operative scores was 1.66 (CI 1.25–2.07). Anterior cingulotomy (two studies, *n =* 22) and anterior capsulotomy (three studies, *n* = 42) showed similar effect sizes: 1.51 (CI 0.82–2.20) vs. 1.74 (CI 1.23–2.26). Multiple procedures were performed in 30 of 116 (25.9%) patients. Based on patient-level data, 53% (*n* = 47) were responders (≥50% improvement), of which 34% reached remission (MADRS ≤ 10 or BDI ≤ 11). BDI mean improvement was 16.7 (44.0%) after a second procedure (*n* = 19). (4) Conclusions: The results are supportive of the benefit of SRA in selected patients with refractory MDD.

## 1. Introduction

Major depressive disorder (MDD) is one of the most common mental disorders, causing significant global disability, morbidity, and mortality with more than 163 million affected worldwide [1]. The aetiology and pathophysiology of MDD remain uncertain. However, neuroimaging, lesion analysis, and post-mortem studies implicate a range of cortical and subcortical structures, including the limbic system, hippocampus and amygdala, and the medial prefrontal cortex, which is formed by parts of the cingulate gyrus and orbitofrontal cortex [2].

Although most patients respond to a combination of standard therapies (psychotherapy, pharmacotherapy, and, in more severe cases, electroconvulsive therapy), up to one-third may not respond adequately [3] and suffer from treatment-refractory depression (TRD). These patients have a less favourable prognosis, and a large proportion still has symptoms two or more years after illness onset [4]. Comorbidity with anxiety disorders such as generalised anxiety disorder is common and further affects outcome [5,6].

MDD is commonly treated by a combination of psychotherapy and medication, with absence of depressive symptoms being the therapeutic goal. When standard treatments fail to show sufficient benefit, neurosurgery for mental disorders may be considered [5]. Contemporary stereotactic radiofrequency ablation (SRA) is very different from historical procedures and enables minimally invasive targeting of deep brain structures by placing the brain within a fixed frame of reference while using a specific coordinate system to define any point in the brain in three dimensions (Figure 1A) [7]. Magnetic resonance imaging (MRI) allows direct visualisation of individual neuroanatomy, permitting safe and accurate lesioning of specific targets [8]. A radiofrequency probe is advanced through the brain to the target, where a high-frequency electrical current is passed through the uninsulated tip. Agitation of ions within the tissue results in frictional heating. The degree of tissue coagulation is controlled by monitoring the temperature in the electrode tip [9]. This approach can disrupt networks that are presumed to be dysfunctional and improve associated symptoms in both movement and mental disorders.

Today, the two most used SRA procedures for MDD are anterior capsulotomy (ACAPS) and anterior cingulotomy (ACING) (Figure 1B,C). Stereotactic sub-caudate tractotomy (SCT) and limbic leukotomy—combining SCT with ACING are much less commonly performed. All procedures are usually performed bilaterally under either local or general anaesthesia.

Suitability for surgery is carefully assessed by a multidisciplinary team including both psychiatrists and neurosurgeons. Patients must meet established criteria for MDD with documented evidence that symptoms are refractory to multiple types of non-surgical treatments. Surgical contraindications typically include ongoing substance misuse, and serious underlying health conditions are also considered [10]. Procedures are only performed with the patient’s informed consent, and in most cases, surgery can only proceed within a strict legal and governance framework. Typical inclusion and exclusion criteria are listed in Appendix A although there is often some variation between centres with regards to minimum age, duration of illness, and specific psychiatric comorbidities. Following surgery, psychotherapy, pharmacotherapy, and follow-up care are essential since the full benefit of stereotactic ablation may not be seen for many months or even years.

Published guidelines from the World Society for Stereotactic and Functional Neurosurgery (WSSFN) state that “stereotactic ablative procedures do not have level I evidence … but their safety and efficacy are supported by level II evidence in treatment-refractory major depressive disorder” [11]. Numerous narrative and systematic reviews have been published [12,13,14], but these rarely attempt to evaluate the quality of the evidence or provide a synthesis of key findings. Whilst we recognise the issues from applying traditional meta-analytic approaches to observational studies, it is unlikely that large, randomised trials of SRA will ever be conducted. Further, attempts at meta-analysis of observational studies are becoming increasingly common and go beyond traditional integrated reviews [15]. Since meta-analyses of observational studies can be undertaken [16], and given the uncertainty about the effectiveness of SRA for TRD, our aims were to: (1) summarise outcomes from SRA studies that met specific criteria; (2) report on adverse effects; (3) compare the results of ACING and ACAPS; and (4) provide recommendations for clinicians whilst being mindful of the limited evidence.

## 2. Materials and Methods

The research protocol was registered on PROSPERO (CRD42020197885) before performing the systematic review.

### 2.1. Inclusion Criteria

Inclusion criteria were:(1)The intervention had to be one of anterior cingulotomy or anterior capsulotomy. Studies that included multiple or combinations of these treatments were included if outcomes from single procedures were available. Where outcomes were for multiple procedures, these patients were not included.(2)The surgical indication was depressive illness. Studies that reported on depressive symptoms in the context of other primary diagnoses were excluded.(3)Measures of depressive symptoms were reported using validated scales at baseline and at least six months after surgery.(4)The study reported outcomes for at least eight patients, reducing risk of statistical anomalies during meta-analysis.

### 2.2. Information Sources and Search Strategy

This systematic review and meta-analysis followed the Preferred Reporting Items for Systematic Reviews and Meta-analyses guidelines [17] (PRISMA; Appendix B). Four electronic databases were searched (*PubMed*, *Embase*, *Web of Science*, and *PsycINFO*) to identify relevant studies on SRA for TRD published from database inception to 1 September 2022. The same keywords were used for each database search and included: “anterior capsulotomy AND (refractory) depression”, “anterior cingulotomy AND (refractory) depression”, “limbic leukotomy/leucotomy AND (refractory) depression”, “subcaudate tractotomy AND (refractory) depression”, “stereotactic ablative (neuro)surgery AND (refractory) depression”, and “stereotactic ablative (neuro)surgery AND major depressive disorder”. Bibliographical database searches were supplemented by hand-searching citations and reference lists of relevant articles and previous systematic reviews [10,18,19,20,21,22,23]. Articles were restricted to English. Authors from the selected papers were contacted for Appendix A, which was provided for three of the studies [18,19,20].

### 2.3. Study Selection and Data Extraction

Data were extracted by three researchers (P.M., L.Z., D.C.) and recorded on a spreadsheet that included information on study characteristics (e.g., publication year, design, and patient numbers), demographic details (e.g., age and sex), symptom ratings (e.g., baseline, post-operative scores, length of follow-up), and adverse events. Data for patients undergoing multiple SRA procedures were extracted separately, with the primary outcome being change after the first procedure only. Studies from the same institution were examined to avoid duplicate reporting of outcomes.

### 2.4. Primary Outcomes

The meta-analysis primary outcome was the change in validated depression score following a single SRA procedure. Data for patients who had undergone multiple procedures were also collected, but these were excluded from the primary analysis if outcome data were not available after the first SRA procedure. In order to account for different rating scales being reported, the percentage difference after surgery was changed to standardised mean difference (effect size) [24].

### 2.5. Data Analysis

Summary data were collated using Microsoft Excel, and meta-analysis was conducted using RevMan 5.4.1 [25]. Risk of bias was assessed using the Cochrane risk of bias tool. A random-effects model was applied. Outcomes were presented using forest plots. Heterogeneity was assessed using a chi-square test and the I^2^ statistic [26].

### 2.6. Patient Level Analysis

Due to the small number of trials and the variation in patient numbers between studies, we also conducted a patient-level analysis of results. The pre- and post-operative BDI (three studies) and MADRS scores (two studies) were recorded separately for each patient. When incomplete, scores were imputed using either the group baseline mean for missing baseline scores or the last observation carried forward (LOCF) method for missing post-operational scores; this is a common and conservative statistical approach with missing follow-up data [27].

Response rates were also calculated at individual patient level, with “response” being defined by ≥50% decrease in depression scores from baseline [28] and “partial response” a reduction of 35–49% [29]. “Remission” was defined as post-treatment MADRS ≤ 10 [30] or BDI ≤ 11 [31]. Deterioration was defined as any depression scores increase from baseline to follow-up. The prevalence of commonly reported adverse effects was estimated based on rates reported in the studies.

### 2.7. Role of the Funding Source

There was no specific award or grant for this study. P.M., D.C. and L.Z. had full access to the data, and all authors had final responsibility for the decision to submit for publication.

## 3. Results

The study selection is shown in Figure 2. The search strategy identified 126 unique records for screening, with 98 records excluded based on information in the abstract. Twenty-eight studies reported on SRA in MDD and were assessed for eligibility. In total, five studies (three prospective [10,18,19] and two retrospective [20,21]) were included in the final analysis. Of the included studies, two reported outcomes following ACING [10,18] and three following ACAPS [19,20,21].

The most-commonly used depression scales were the BDI (self-reported) and the MADRS (clinician-rated). All studies had missing data either at baseline or post surgery. Reasons included different scales being used at different timepoints, studies reporting multiple procedures but not outcomes from the first procedure, and incomplete follow-up. It is unlikely that data are missing completely at random. The MADRS had the most complete data. Outcomes for anxiety were available in some studies (using the Beck Anxiety Inventory (BAI) or the Hospital Anxiety and Depression Scale (HADS)), but data were not comprehensive enough to permit meta-analysis.

### 3.1. Risk of Bias

All included studies had a high risk of bias, mainly arising from their non-randomised nature, lack of control groups, and outcome assessment by the treating clinical teams. Many studies had missing follow-up data, possibly arising from the long duration of follow-up for some patients. A risk of bias summary for included studies is shown in Figure 3.

### 3.2. Meta-Analysis: Reported Data Only

Based on complete individual patient data, an effect size of 1.66 (95% confidence interval (CI), 1.25–2.07) for SRA surgery (comparing post-operative vs. pre-operative scores) was found, synthesising the five trials using a random-effects model (Figure 4). No statistical evidence for effect size heterogeneity was found (*χ*^2^ = 0.28, df = 4, *p* = 0.92; I^2^ = 0%). Despite lack of statistical heterogeneity, a random-effects model was used because of the variation in depression scales.

The combined effect size for the three ACAPS studies was 1.74 (95%CI 1.23–2.26). There was no evidence for effect size heterogeneity (*χ*^2^ = 0.64, df = 2, *p* = 0.72; I^2^ = 0%). The two ACING studies reported a similar response, with an effect size of 1.51 (95%CI 0.82–2.20). No evidence for effect size heterogeneity was found (*χ*^2^ = 0.00, df = 1, *p* = 0.96; I^2^ = 0%).

### 3.3. Meta-Analysis: LOCF Analysis

The meta-analysis showed similar results when LOCF was used to replace missing data. Using a random-effects model, an effect size of 1.41(95%CI 1.06–1.76) for SRA surgery was found (Figure 5). There was no evidence for effect size heterogeneity (*χ*^2^ = 0.10, df =1, *p* = 0.76; I^2^ = 0%).

The three ACAPS studies displayed an effect size of 1.38 (95%CI 0.96–1.81), with no evidence for effect size heterogeneity (*χ*^2^ = 2.14, df = 2, *p* = 0.34; I^2^ = 7%). The two ACING studies reported a similar response, with an effect size of 1.51 (95%CI 0.82–2.20). Again, no evidence for effect size heterogeneity was found (*χ*^2^ = 0.00, df = 1, *p* = 0.96; I^2^ = 0%).

### 3.4. Patient-Level Analysis

The total number of patients reported in the trials was 116 (43 = male; 73 = female; mean age 43.8 years, range 21–69), but only 64 met inclusion criteria for the meta-analysis primary outcome (ACING *n =* 22; ACAPS *n =* 42; Table 1). Adverse event data were available for 108 patients since one paper did not report on adverse events. A total of 30 patients underwent multiple procedures.

Based on complete patient-level data (*n* = 47), 53% of patients were responders, 34% met criteria for remission, 11% partially responded, and a further 26% had some improvement in baseline depressive symptoms. Response rates are displayed in Appendix A.

Data for the mean percentage change in anxiety scores were available for three papers (including additional data from Dr. David Linden, personal communication). All studies showed similar improvements in anxiety scores (measured using the HADS) pre- to post ablative surgery (ACING 47.6%, *n =* 8 [18]; ACAPS 42.9%, *n =* 5) [19]. Using the BAI, changes were similar (ACAPS 45.1%, *n =* 13 [20]), with a weighted mean improvement across all studies of 45.4%. However, data were incomplete, and it is not known if changes in anxiety were independent of improvements in mood.

Out of 116 patients in the included studies, 30 (25.9%) underwent multiple SRA procedures. Outcomes were available for 19 of these patients, with a mean improvement in BDI score of 16.7 (44.0% reduction) after a second SRA procedure.

### 3.5. Adverse Effects

All but one of the selected papers reported on adverse effects that occurred after SRA for TRD. The most frequently reported side effects are listed in Table 2. Short-term adverse effects were defined as side effects that were experienced immediately after surgery but resolved within one year. Long-term adverse effects were defined as those that persisted beyond one year.

The most common short-term adverse events reported after ACAPS were urinary incontinence (41%); confusion and disorientation (43%); fatigue (23%); headache (12%); and memory problems (9%). Short-term urinary incontinence was also reported following ACING but at lower rates. Most adverse effects were transient, disappearing within a few weeks after surgery and usually resolving within a year. However, decrease in motivation and memory and concentration difficulties lasted longer than 12 months for some patients, mostly after ACAPS. MDD is characterised by lower motivation, energy, and ability to concentrate. Moreover, fatigue, headache, and weight gain are relatively common symptoms in patients with severe depression regardless of treatment. In the absence of suitable control groups, it is not clear whether these symptoms arose from SRA or whether they were part of the pre-existing depressive disorder.

One paper mentioned pre-operative suicidal ideation in eight patients [20], and only one paper reported on change in suicidal ideation, stating improvement in all ten patients at 12 months [21]. Attempted suicide rates prior to surgery are mentioned in two papers, with a history of attempted suicide in 23 of 37 patients (62.2%) [19,20], but attempted suicide rates after surgery are not reported in any paper. Although no completed suicides were reported after surgery, the numbers were too small to draw firm conclusions on surgical effects on suicidal ideation.

Although not reported in detail here, 4/5 included studies [18,19,20,21] compared neuropsychology battery test results before and after surgery in a subset of patients. All studies concluded that neurocognitive and personality testing were not significantly different at follow-up. Three studies reported a trend towards improvement on some measures of executive function.

## 4. Discussion

### 4.1. Summary of Main Findings

The available data suggest that SRA is an effective therapy that offers a meaningful chance of improvement. This meta-analysis attempted to synthesise the change in depression symptom scores following SRA for MDD. The effect size for surgery using complete individual data was 1.66 (95%CI 1.25−2.07), with comparable effect sizes after ACING and ACAPS (1.51 and 1.74, respectively). Similar effect sizes were seen when missing data were imputed using LOCF (mean 1.41), once again with no significant differences between procedures: ACING, 1.51; ACAPS, 1.38. These improvements represent a large and potentially clinically relevant improvement in symptom scores, with a transition from “severe” to “low moderate” on the BDI and from “severe” to “mild” depression on the MADRS [32,33].

A significant number of individuals (25.9%) had a second procedure due to an unsatisfactory response to the first surgery. In this patient group, BDI scores improved by a mean of 16.7 (44.0%) after a second SRA procedure (*n =* 19). Multiple procedures seem to be beneficial in most patients even when the initial intervention failed to achieve satisfactory results.

SRA for MDD appears to be relatively safe. Surgical mortality and suicide have not been reported. Many long-term adverse events, such as lower motivation, energy, ability to concentrate, fatigue, and headache, are commonly found in patients with severe depression regardless of treatment. Moreover, neurocognitive and personality testing were not negatively affected.

### 4.2. GRADE Recommendation

The strength of recommendation, based on the GRADE system [34] is “weak” based primarily on the quality of the underlying evidence but also the low numbers of studies, small numbers of participants, and persisting uncertainty between desirable and undesirable effects. Consequently, a cautious approach to evidence appraisal was adopted despite an apparently large effect size. Larger, well-controlled studies are likely to influence effect size and, possibly, the direction. Consistency of reporting of patient characteristics, clinical outcomes, and adverse effects as per Nuttin et al. [11] is highly desirable.

However, some context for this grading is important. First, in this patient population, the evidence for any treatment beyond the first few antidepressant trials is weak, large numbers of trials to guide treatment decisions are absent, and we lack effective treatments for patients with severe and chronic depression. Second, despite recognisable uncertainty about both frequency and severity of adverse effects, consistently high rates of serious adverse effects were not reported. Third, although estimated effect size is variable (a likely consequence of small sample sizes), the direction of effect is consistent. Many patients experience measurable improvements in symptoms without experiencing high rates of harmful effects. This should provide reassurance to clinicians when considering further management of treatment refractory patients.

### 4.3. Study Strengths

When meta-analysing observational data, there are several sources of bias. Although bias cannot be eliminated, it has been assessed and reported in detail. Further, when follow-up is conducted over long periods of time, missing data are inevitable. We have tried to address this by reporting findings for complete samples only. Reported response rates for procedures are based on individual patient-level data. Moreover, where missing data were imputed, a conservative approach was used: LOCF.

### 4.4. Study Limitations

This study has several limitations. First, data are all from uncontrolled, open-label studies, and therefore, it is not possible to conclude with certainty that the surgical intervention was directly responsible for symptomatic change. Second, there were missing data. Since pre-operative and post-operative scores were not available for all patients, there is the possibility that available patients were not representative of all individuals receiving the intervention. Studies often try to overcome this problem by using imputed data, but this is not always possible when sample sizes are small. Authors were contacted to obtain individual patient data, missing baseline scores were replaced with the group baseline mean, and LOCF was used for missing post-operational scores. Nevertheless, significant data gaps limit the validity of this meta-analytical approach, as listed in Appendix A. Third, studies used different rating scales to measure outcomes. Although effect sizes were used to pool study data, reported outcomes combined self-reported with clinician-rated outcomes. Fourth, although suicidality is common in this patient population, it was not reported in detail in any of the papers. Fifth, the studies included in this meta-analysis assessed patients at different times after surgery but did not always report systematic follow-up over multiple timepoints. Since MDD is a chronic and relapsing-remitting disorder, these timed assessments may give a distorted view about long-term well-being. Finally, although patient numbers in each study were small, all studies reported outcomes that were positive, and there was no clear evidence of a systematic approach to the collection of data on adverse effects.

### 4.5. Comparison to Deep Brain Stimulation

Over the past 20 years, deep brain stimulation (DBS) has superseded (but not replaced) stereotactic ablation in the management of treatment refractory movement disorders such as Parkinson’s disease, dystonia, and tremor. This development has sparked an interest in the use of DBS for mental disorders. A recently published review and meta-analysis reported on response and remission rates after open-label studies of DBS for TRD at different anatomical targets. Comparing the outcome of SRA with those of DBS for TRD revealed very similar response (53% vs. 56%) and remission rates (34% vs. 35%) [35]. Further, since DBS surgery is followed by multiple follow-up programming sessions, non-specific treatment effects of DBS are likely to be greater than that of stereotactic ablation. Combined with DBS usually being higher cost, with the need for significant additional clinical infrastructure, and higher risks of infection, the perception that DBS offers superior clinical advantage over SRA may be incorrect. Nevertheless, the irreversibility of ablative surgery and the theoretical risk of permanent neuropsychological impairment requires robust patient selection and a rigorous informed consent process [36].

### 4.6. The Future of SRA for TRD

Whilst this study focused on radiofrequency ablation, other methods of lesion generation (for example gamma-knife surgery and MRI-guided focus ultrasound) are available. These methods are “incisionless” and use gamma radiation or ultrasound (respectively) to create a lesion at the target point. Results of their use to perform ACAPS or SCT in patients with MDD are encouraging but are limited to case reports and one small phase I trial [37,38]. However, these technologies may come to be seen as more acceptable by some patients and psychiatrists.

Prospective, double-blind, randomised, sham-controlled studies are the usual “gold standard” to objectively assess the effectiveness of surgical procedures. However, such trials are difficult to undertake and face several ethical and logistical challenges [39]. Failing this, large, open-label studies with complete data collection, preferably with a control arm allocated to best non-surgical treatment, are desirable. It is important that future publications on SRA for MDD ensure comprehensive data collection at specific timepoints, using standardised scales and preferably including both self- and clinician-rated scales. Comprehensive data on suicidality, anxiety symptoms, details regarding the use of multiple procedures, and comprehensive information regarding adverse events should be reported for all patients undergoing SRA. Ideally, individual patient data should be provided (in Appendix A if necessary) to allow improvements in depression and anxiety scores after SRA to be matched to the experience of adverse events. Long-term, preferably life-long, follow-up of patients is desirable to fully evaluate outcomes of SRA in MDD.

Stereotactic ablation for TRD remains limited to few centres around the world, often with small numbers of patients referred for treatment. Given the high mortality, morbidity, and burden on society of MDD, the paucity of high-quality outcome reporting for stereotactic radiofrequency ablation in the last two decades is notable. However, the data presented here suggest that SRA is a promising therapy in this patient group and is likely to offer a meaningful chance of improvement. Referral of larger numbers of patients with TRD for consideration of stereotactic ablation may allow centres to design studies that will allow us to better understand the role of SRA in the management of patients who have not responded to all other available treatments. This must occur in the context of an experienced multidisciplinary team, within a framework of strong clinical governance and safeguarding.

## Figures and Tables

**Figure 1 brainsci-12-01379-f001:**
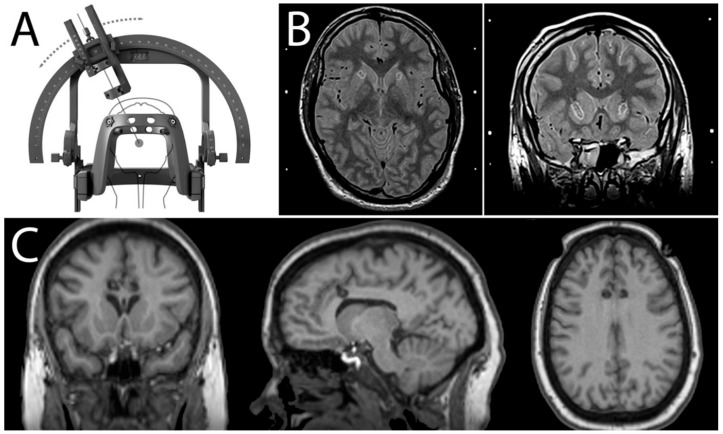
(**A**) Stereotactic frames use a specific coordinate system to define and provide surgical access to any point in the brain in three-dimensional space (Leksell^®^ VantageTM Stereotactic System-image courtesy of Elekta). (**B**) Axial and coronal stereotactic proton-density-weighted MR images immediately after anterior capsulotomy. The lesions can be seen as a hypointense area of tissue necrosis surrounded by a hyperintense region of oedema. (**C**) Coronal, sagittal, and axial stereotactic T1-weighted MR images immediately after anterior cingulotomy. (Fiducial markers have been cropped out of the original images.) The lesions can be seen as a hypointense area.

**Figure 2 brainsci-12-01379-f002:**
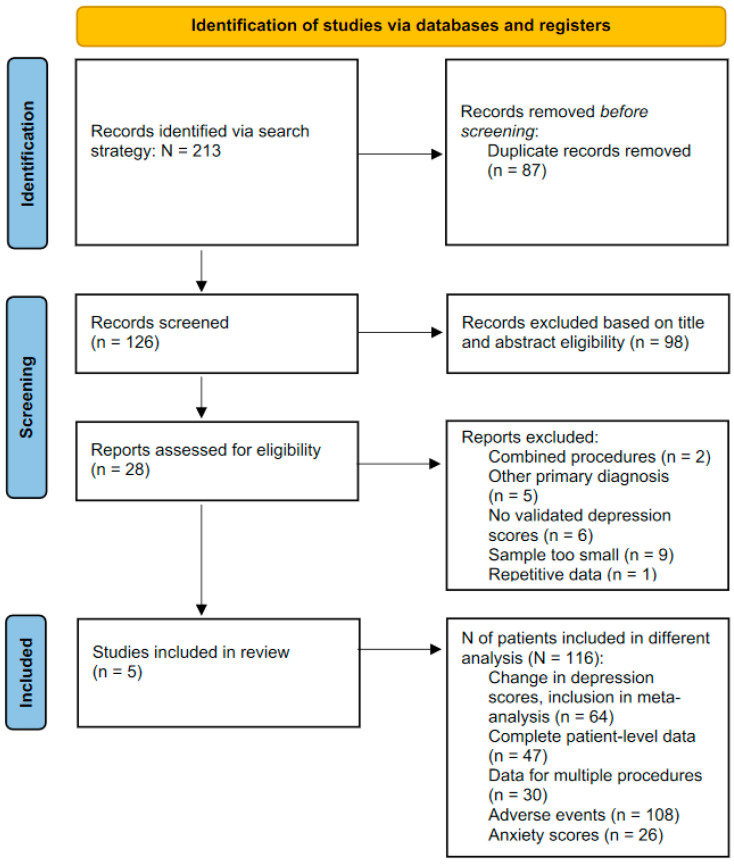
PRISMA flow diagram summarising the study selection process.

**Figure 3 brainsci-12-01379-f003:**
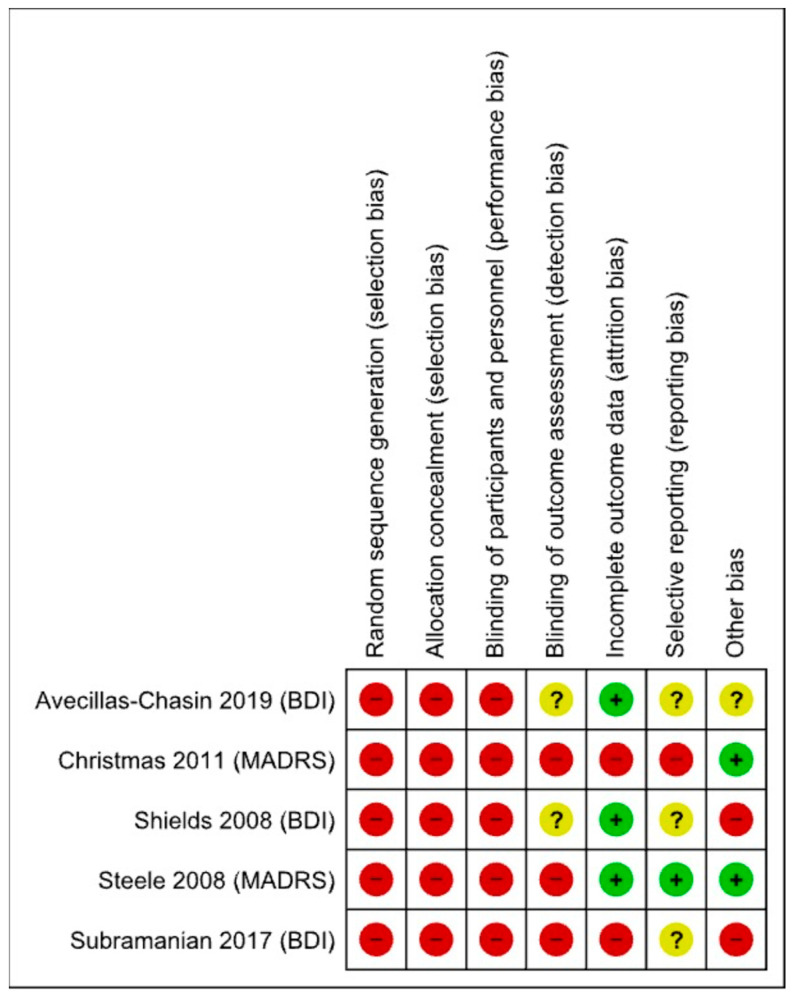
Risk of bias summary for included studies, using the Cochrane risk of bias tool. Red = High risk; Green = Low risk; Yellow = Unclear. [10,18,19,20,21].

**Figure 4 brainsci-12-01379-f004:**
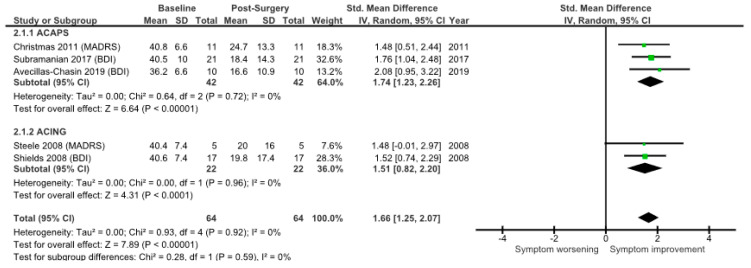
Standardised mean difference in depression scores pre- and post-ablative surgery for ACAPS and ACING. Only patients where individual patient data are available are included. The mean change for each study is represented by a green box. More powerful studies are indicated by a larger sized box, and they contribute to the pooled result to a greater degree [10,18,19,20,21].

**Figure 5 brainsci-12-01379-f005:**
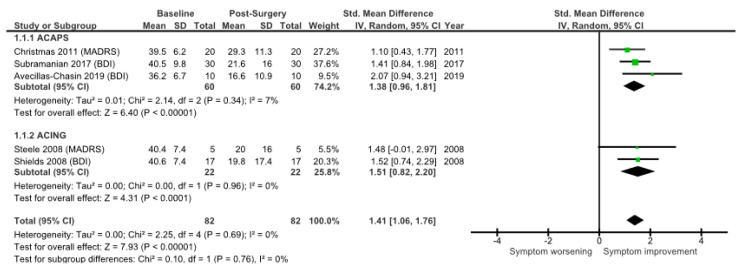
Standardised mean difference in depression scores pre- and post-ablative surgery for ACAPS and ACING. All patients (single procedure and LOCF) included. The mean change for each study is represented by a green box. More powerful studies are indicated by a larger sized box, and they contribute to the pooled result to a greater degree [10,18,19,20,21].

**Table 1 brainsci-12-01379-t001:** Included studies and patients (individual patient data only). N, number of patients; %, proportion of patients included in review. Data in bold represents pooled data from more than one study.

Procedure and Study	N	%	Follow up Timepoint/Months	Primary Outcome
ACAPS	42	65.6		
Avecillas-Chasin et al. [21]	10	15.6	12	BDI
Subramanian et al. [20]	21	32.8	6 (median)	BDI
Christmas et al. [19]	11	17.2	12	MADRS
**ACING**	**22**	**34.4**		
Shields et al. [10]	17	26.6	30 (mean)	BDI
Steele et al. [18]	5	7.8	12	MADRS
**Total**	**64**	**100**		

**Table 2 brainsci-12-01379-t002:** Most frequently reported adverse events across all selected studies (*n =* 108). Short-term is defined as persisting less than 12 months, whereas long-term as persisting more than 12 months. * No data regarding adverse events were available for Steele et al. [18].

	Anterior Capsulotomy		Anterior Cingulotomy	
Adverse Events *	Short-Term (%)	Long-Term (%)	Short-Term (%)	Long-Term (%)
Confusion/disorientation	42.7	-	-	-
Urinary incontinence	41.3	4.0	12.1	-
Fatigue	22.7	4.0	-	-
Headache	12.0	4.0	-	-
Memory problems	9.3	13.3	-	3.0
Apathy	5.3	14.7	-	-
Concentration/attention impairment	5.3	10.7	-	-
Motor weakness	4.0	-	-	-
Weight gain	2.7	5.3	-	-
Infection	2.7	-	3.0	-
Seizures	1.3	2.7	-	3.0
Personality change	-	5.3	-	-

## Data Availability

Data collected for the study are available upon reasonable request.

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
