# Peer review of "Stereotactic Radiofrequency Ablation for Treatment-Refractory Depression: A Systematic Review and Meta-Analysis"

_brainsci, 2022, doi:10.3390/brainsci12101379_

Round 1

Reviewer 1 Report

The manuscript titled "Stereotactic radiofrequency ablation for treatment-refractory depression: a systematic review and meta-analysis" is interesting. Although it covers an important topic, some improvement might be needed. 

Minor issues 

(1) Figures 2 and 3 do not have any descriptions. 

(2) There are various grammatical errors and typos; such as putting the reference after the full stop. 

Major issues:

(3) The introduction section is limited in its coverage of the need to use this particular method in treating depression; why clinicians may resort to performing neurosurgery. What are the current recommendations regarding its usage? 

(4) The surgical method itself is only briefly described. A more in-depth explanation of the procedure is required. 

(5) The part of the discussion covering the interpretation of the results is also limited. It is not clear whether the method is effective, efficient, or safe. What are the downsides of using it? 

(6) The conclusion is not clear. Would the authors recommend using this method? Are the results significant? In which cases, if any, could this type of neurosurgery be needed? 

(7) The authors mentioned that repeating the procedure could be beneficial, but what are the conditions for this? What type of patient would benefit from this procedure? 

(8) Did the study take into consideration comorbidity?

Reviewer 2 Report

I consider that the review entitled "Stereotactic radiofrequency ablation for treatment-refractory depression: a systematic review and meta-analysis” is ready for publication. I only comment that the discussion section seems to be written in a smaller font size than the rest of the manuscript

Reviewer 3 Report

Herein, a systematic literature review and meta-analysis on stereotactic radiofrequency ablation for Major Depressive Disorder is presented. The authors included specific electronic databases to identify relevant studies on stereotactic radiofrequency for TRD published from database inception to September 1, 2022. The search strategy identified 126 unique records for screening, with 98 records excluded based on information in the abstract. The study summarize outcomes from SRA studies that met specific criteria and upon adverse effects account and comparison between the results of ACING and ACAPS which are the two most used SRA procedures for MDD, recommendations for clinicians whilst being mindful of the limited evidence are accommodated. In general, the provided information is satisfactory. However, I have some few comments and suggestions:

1.       Why they authors stayed on these four inclusion criteria? A reference should be given

2.       Apart from the LOCF method, could you suggest other approach for missing post-operational scores or not?

3.       The authors suggest the BDI and the MADRS (self-reported and clinician-rated, respectively) as the most-commonly used depression scales. What about the Hamilton Depression Rating Scale or other?

4.       Can the authors clarify how they concluded to the Risk of bias for included studies (protocol, methodology?)

5.       The quality of Figures 4 and 5 can be improved

6.       In Table 1, please indicate the meaning of N and %. References should be given as numbers.

7.       Figure 1 is original? Otherwise, a reference should be given

Round 2

Reviewer 1 Report

I have no further comments.